

# A new spatiotemporal two-stage standardized weighted procedure for regional drought analysis

Rizwan Niaz[1], Nouman Iqbal[1,2], Nadhir Al-Ansari[3], Ijaz Hussain[1], Elsayed Elsherbini Elashkar[4], Sadaf Shamshoddin Soudagar[5], Showkat Hussain Gani[6], Alaa Mohamd Shoukry[7,8] and Saad Sh. Sammen[9]

[1] Statistics, Quaid-i-Azam University, Islamabad, Punjab, Pakistan
[2] Knowledge unit of business Economics accountancy and Commerce (KUBEAC), University of management and technology Sialkot campus, Sialkot, Pakistan
[3] Department of Civil, Environmental and Natural Resources Engineering, Lulea University of Technology, Lulea, Sweden
[4] Administrative Sciences Department, Community College, Riyadh, Riyadh, Saudi Arabia
[5] College of Business Administration, King Saud University Riyadh, Riyadh, Saudi Arabia, Riyadh, Saudi Arabia
[6] Business Administration, College of Business Administration, King Saud University Riyadh, Saudi Arabia, Riyadh, Riyadh, Saudi Arabia
[7] Arriyadh Community College, King Saud University, Riyadh, Saudi Arabia
[8] Workers University, KSA, Nsar, Egypt, Egypt
[9] Department of Civil Engineering, Coolege of Engineering, University of Diyala, Diyala Governorate, Iraq

Corresponding authors
Rizwan Niaz,
rizwanniaz@stat.qau.edu.pk
Alaa Mohamd Shoukry,
aabdulhamid@ksu.edu.sa

## ABSTRACT

Drought is a complex phenomenon that occurs due to insufficient precipitation. It does not have immediate effects, but sustained drought can affect the hydrological, agriculture, economic sectors of the country. Therefore, there is a need for efficient methods and techniques that properly determine drought and its effects. Considering the significance and importance of drought monitoring methodologies, a new drought assessment procedure is proposed in the current study, known as the Maximum Spatio-Temporal Two-Stage Standardized Weighted Index (MSTTSSWI). The proposed MSTTSSWI is based on the weighting scheme, known as the Spatio-Temporal Two-Stage Standardized Weighting Scheme (STTSSWS). The potential of the weighting scheme is based on the Standardized Precipitation Index (SPI), Standardized Precipitation Evapotranspiration Index (SPEI), and the steady-state probabilities. Further, the STTSSWS computes spatiotemporal weights in two stages for various drought categories and stations. In the first stage of the STTSSWS, the SPI, SPEI, and the steady-state probabilities are calculated for each station at a 1-month time scale to assign weights for varying drought categories. However, in the second stage, these weights are further propagated based on spatiotemporal characteristics to obtain new weights for the various drought categories in the selected region. The STTSSWS is applied to the six meteorological stations of the Northern area, Pakistan. Moreover, the spatiotemporal weights obtained from STTSSWS are used to calculate MSTTSSWI for regional drought characterization. The MSTTSSWI may accurately provide regional spatiotemporal characteristics for the drought in the selected region and motivates researchers and

policymakers to use the more comprehensive and accurate spatiotemporal characterization of drought in the selected region.

# INTRODUCTION

Drought is a creeping phenomenon, influencing more individuals than other natural hazards (*Botai et al., 2017*; *Mousavikhah, Shayegh & Ahmadpari, 2020*; *Bhunia, Das & Maiti, 2020*; *Elhoussaoui, Zaagane & Benaabidate, 2021*). It is a slowly evolving and multifaceted disaster, often poorly understood in the perspective of regional climatic, hydrological, and human environment (*Mukherjee, Mishra & Trenberth, 2018*; *Vicente-Serrano et al., 2020*). Drought is a recurring natural hazard appearing in all climatic zones worldwide and significantly influences social and economic well-being, ecological environment, and agricultural sectors (*Guneralp, Guneralp & Liu, 2015*; *Hagenlocher et al., 2019*; *Shah & Mishra, 2020*). In its explicit form, drought can be described as the water discrepancy that appears in several types, including agricultural, meteorological, hydrological, and socio-economic drought (*Lai et al., 2019*; *Jiang & Wang, 2019*; *Vernieuwe, De Baets & Verhoest, 2020*; *Diaz et al., 2020a*). The meteorological drought occurs due to insufficient precipitation, and insufficiency in soil water supply triggers the agricultural drought. If drought distribution further continues *via* the hydrological cycle, a deficiency in surface or groundwater evolves, causing hydrological drought (*Vernieuwe, De Baets & Verhoest, 2020*; *Diaz et al., 2020b*).

Further, the monitoring, modeling, and prediction of meteorological droughts are of utmost significance (*Mazhar et al., 2020*; *Guneralp, Guneralp & Liu, 2015*; *Hagenlocher et al., 2019*; *Jiang & Wang, 2019*). Because the meteorological drought becomes the root for other drought types due to insufficient precipitation (*Wu et al., 2017*; *Lai et al., 2019*; *West, Quinn & Horswell, 2019*; *Vernieuwe, De Baets & Verhoest, 2020*). Moreover, accurate evaluation of meteorological drought brings useful information for decision-makers worldwide working in several fields associated with agriculture, hydrology, industrial, and water-budget managers to formulate precautionary measures and develop future planning (*Alizadeh & Nikoo, 2018*). Further, various studies have described the importance of monitoring regional drought (*Wilhite, 2000*; *Zhai & Feng, 2009*; *Zhang et al., 2012*; *Van Lanen et al., 2016*; *Santos et al., 2019*). The regional monitoring of drought has significantly influenced the country's economy and other human activities (*Zhai & Feng, 2009*; *Mousavikhah, Shayegh & Ahmadpari, 2020*). The regional drought monitoring highlights those issues if they could improve at the regional level before the events occur, then potential adverse effects of drought can be minimized in the future (*Wilhite, 2000*; *Santos et al., 2019*; *Pontes Filho et al., 2019*; *Pontes Filho et al., 2020*). More comprehensive and accurate drought monitoring can be possible by applying suitable tools and techniques according to climatic conditions. Based on the various climatic conditions, several drought indices have been used for drought monitoring. These indices require proper and effective recording

related to drought occurrences. For instance, more accurate estimation of drought indices requires appropriate gauge stations with suitable records for regional drought.

Information obtained from the drought indicators can be used for improving drought predicting and forecasting (*Tsakiris, Pangalou & Vangelis, 2007*; *Niemeyer, 2008*; *Hayes et al., 2011*; *Mukherjee, Mishra & Trenberth, 2018*). Moreover, various studies from the literature have discussed the drought indices. Several studies have developed some new drought indices (*McKee, Doesken & Kleist, 1993*; *Tsakiris, Pangalou & Vangelis, 2007*; *Vicente-Serrano, Beguería & López-Moreno, 2010*). The development in drought monitoring is leading to enhancing the capabilities of drought monitoring more precisely and accurately. The standardized indices are used for the drought classifications. The estimation of the drought indices is based on the various parameters (precipitation, temperature, etc.). However, the preference for calculating the indices is based on climatic conditions of the available data (*Niemeyer, 2008*; *Hayes et al., 2011*). For instance, an index based on the precipitation, which is known as Standardized Precipitation Index (SPI) proposed by *McKee, Doesken & Kleist, 1993*, the Reconnaissance Drought Index (RDI) developed by *Tsakiris, Pangalou & Vangelis (2007)*, the *Vicente-Serrano, Beguería & López-Moreno (2010)* has proposed an index which is called as Standardized Precipitation Evapotranspiration Index (SPEI).

Further, knowledge about the spatiotemporal characteristics of drought occurrences is crucial for drought monitoring and mitigation policies (*Maybank et al., 1995*; *SIRDAŞ & Sen, 2003*; *Wang et al., 2020*; *Zhou et al., 2020*). Specifically, in the presence of a homogenous region, the indicators need comprehensive characterization of the drought that provides regional spatiotemporal information. Information obtained from the spatiotemporal characteristics can be used for significant, drought monitoring, modeling, and prediction (*Maybank et al., 1995*; *SIRDAŞ & Sen, 2003*; *Corzo Perez et al., 2011*; *Wang et al., 2020*; *Diaz et al., 2020b*). Therefore, an intense spatiotemporal procedure is required to assimilate the spatiotemporal information of the selected homogenous region (*Dabanlı, Mishra & Şen, 2017*; *Caloiero et al., 2018*; *Zhou et al., 2020*). In this regard, we aimed to develop a new drought assessment procedure for the regional drought characterization. The proposed procedure is known as the Maximum Spatio-Temporal Two-Stage Standardized Weighted Index (MSTTSSWI). The MSTTSSWI is based on the Spatio-Temporal Two-Stage Standardized Weighting Scheme (STTSSWS) and validated to the six meteorological stations of the northern area of Pakistan. The proposed MSTTSSWI provides more comprehensive and accurate information about the regional drought characteristics.

## METHODS

### Standardized drought index

The various Standardized Drought Indices (SDI) have been used to monitor drought (*Alley, 1984*; *Narasimhan & Srinivasan, 2005*; *Stagge et al., 2015*; *Eslamian et al., 2017*; *Niaz et al., 2020*). However, the SPI has been commonly used for drought assessment and can be calculated at various time scales. Several studies have used SPI for drought monitoring (*Stagge et al., 2015*; *Eslamian et al., 2017*; *Pathak & Dodamani, 2019*; *Hagenlocher et al.,*

*2019*; *Niaz et al., 2020*). Further, the SPEI is a multi-scalar drought index that attracted significant attraction in drought estimation. The SPEI was developed by *Vicente-Serrano, Beguería & López-Moreno (2010)* that obtains the simplicity in temporal characterization and considers as an extension of SPI. SPEI evaluates the effects of evaporative demand on drought and is computed by considering both precipitation and potential evapotranspiration. More detailed information concerning the SPEI computation can be acquired in *Vicente-Serrano, Beguería & López-Moreno (2010)* and *Beguería et al. (2014)*. Further, both SDI (SPI and SPEI) are frequently used in various studies to assess drought in various regions. The calculation and data availability are relatively easy; therefore, these two indices are commonly used worldwide. Hence, based on the availability of the data, the current study considers both SDI (SPI, and SPEI) for the current analysis. Both SDI requires the appropriate transformation method for standardizing the selected cumulative density function and all the numerical vectors containing the time series data based on the probability plotting formulas. Therefore, following the same procedure is used as a transformation method for standardizing SPI and SPEI (*Farahmand & AghaKouchak, 2015*).

$$SDI = -\left( v - \frac{l_0 + l_1 v + l_2 v^2}{1 + m_0 v + m_1 v^2 + m_2 v^3} \right) \tag{1}$$

For

$$v = \sqrt{\ln\left[ \frac{1}{\{T(x)\}^2} \right]}$$

When
$$0 \le T(x) \le 0.5 \tag{2}$$

$$SDI = +\left( v - \frac{l_0 + l_1 v + l_2 v^2}{1 + m_0 v + m_1 v^2 + m_2 v^3} \right) \tag{3}$$

And for

$$v = \sqrt{\ln\left[ \frac{1}{\{T(x)\}^2} \right]}$$

When

$$0.5 \le T(x) \le 1 \tag{4}$$

where $l_0 = 2.515517$, $l_1 = 0.802853$, $l_2 = 0.010328$, $m_0 = 1.432788$, $m_1 = 0.189269$, $m_2 = 0.001308$.

## The weighting scheme: the spatio-temporal two-stage standardized weighting scheme (STTSSWS)

Drought causes severe damages worldwide. However, drought monitoring policies need a deep knowledge regarding the spatial and temporal distribution of drought risk at the local

or regional level. Therefore, in this perspective, we propose STTSSWS, the innovative methodology giving a better evaluation and management of drought monitoring, especially for spatial and temporal characteristics of the region. The STTSSWS is based on steady-state probabilities. The steady-state probabilities can be defined as the average probability that the system remains in a certain state after many transitions.

Moreover, in a Markov process, it can be more explicitly defined as the probabilities approach the steady-state probabilities after some periods have been passed. Further, detailed mathematical explanations related to the steady-state probabilities of the Markov chain are presented in *Stewart (2009)*. Moreover, in STTSSWS, steady-state probabilities are used as weights in two stages. The application of steady-state probabilities is available (*Niaz et al., 2020*). The *Niaz et al. (2020)* used the steady-state probabilities as a weighting scheme for their studies. They obtained steady states weights from the long run time series data for various drought categories in the selected region. The steady-state probabilities for the drought categories are defined as the visit of the specific drought category in the long run. The steady states probabilities for various drought categories with their probabilities, $w'_{ij}$, $w''_{ij}$ are given as follows:

$$
\begin{array}{ccccccc}
\text{ED} & \text{SD} & \text{MD} & \text{ND} & \text{MW} & \text{SW} & \text{EW}
\end{array}
$$

$$
\text{Steady-state probabilities for SPI} = \begin{bmatrix} w'_{11} & w'_{21} & w'_{31} & w'_{41} & w'_{51} & w'_{61} & w'_{71} \end{bmatrix}
$$

$$
\begin{array}{ccccccc}
\text{EW} & \text{SW} & \text{MW} & \text{ND} & \text{MD} & \text{SD} & \text{ED}
\end{array}
$$

$$
\text{Steady-state probabilities for SPEI} = \begin{bmatrix} w''_{11} & w''_{21} & w''_{31} & w''_{41} & w''_{51} & w''_{61} & w''_{71} \end{bmatrix}
$$

Further, the limiting probability in each index for varying drought categories is a $1 \times 7$-row vector given by the following expressions.

$$
\prod_i (SPI) = \left[ \prod_1 (ED_{SPI}) \prod_2 (SD_{SPI}) \prod_3 (MD_{SPI}) \prod_4 (ND_{SPI}) \prod_5 (MW_{SPI}) \right.
$$
$$
\left. \prod_6 (SW_{SPI}) \prod_7 (EW_{SPI}) \right] \tag{5}
$$

$$
\prod_i (SPEI) = \left[ \prod_1 (ED_{SPEI}) \prod_2 (SD_{SPEI}) \prod_3 (MD_{SPEI}) \prod_4 (ND_{SPEI}) \prod_5 (MW_{SPEI}) \right.
$$
$$
\left. \prod_6 (SW_{SPEI}) \prod_7 (EW_{SPEI}) \right] \tag{6}
$$

Equations (5) and (6) give the long-run probabilities for each drought category for SPI and SPEI, and these probabilities are considered as initial weights for the calculation of STTSSWS. The calculation of STTSSWS is based on two stages. In the first stage, steady states probabilities were used to obtain weights corresponding to each drought category of every month of the time-series data. The monthly time series data are collected from January 1971 to December 2017. Thus, each month appears with a specific drought category. The drought categories are specified by the thresholds values of the drought

indices. Further, the SPI and SPEI used the same thresholds values for the characterization of drought. For example, Normal Dry (ND) appears in any month the threshold for ND is "SDI > −1 & SDI <= 1" for both SDI (SPI and SPEI). The thresholds for other drought categories are defined accordingly. These thresholds for various drought categories are given in intervals. Hence, ND can occur in any month with a specific value of the specified interval. However, we need the particular value (not the interval) as a weight of the ND for any month in the time series data. This need becomes the reason to calculate steady-state probabilities as an initial weighting scheme. The steady-state assign the single probability for ND in the whole data set. This probability is considered as the initial weight for ND. Accordingly, the weights for other drought categories are assigned.

Furthermore, in the second stage of STTSSWS, the weights computed from steady-state probabilities for varying drought categories are used to calculate new spatiotemporal weights. The second stage of STTSSWS is divided into two phases; in the first phase, the weights obtained from steady-state probabilities are being associated with temporal characteristics of the data. For this purpose, firstly, the data of each month from the selected period is combined separately and calculated weights for each month of the year. For example, the ND appears in January 1971 at any station for selected time series data from January 1971 to December 2017; the weight for ND can be calculated using Steady-State Probability (SSP) as initial weights as follows,

$$
\begin{aligned}
&Temporal\ weight_{((ND)\ Janurary\ 1971)} \\
&= \frac{SSP_{(ND)}\ at\ January_{(1971)}}{sum\ of\ SSP\ of\ all\ drought\ categories\ in\ all\ january\ of\ selected\ time\ series\ data}
\end{aligned}
\tag{7}
$$

In Eq. (7) the nominator contains SSP for only ND that is observed in January 1971, however, the denominator term consists of drought categories that appeared in all January (January of 1971, January of 1972, January of 1973, and so on till January of 2017), and their collective sum is computed. Further, we have monthly data of 47 years used at a 1-month time scale (47 * 12 = 564 months), so 47 values of January are included in the data. Accordingly, other months appear 47 times in the selected data set. The formulas are provided only for January of Skardu station to avoid the complication of the mathematical equalities. Further, several drought categories can be observed in 47 years of time series data in Skardu station ("as Eq. (6) presented only for one drought category"). Therefore, a general form is required that defines temporal weights more comprehensively. Equation (8) is provided for the calculation of January at Skardu station. However, the temporal weights for other months in other stations can be calculated based on the same rationale.

$$
T_{January}\left(P_{(mi)(Skardu)}\right) = \frac{W_{(mi)Skardu}}{\sum_{i=1}^{n} W_{(mi)Skardu}}, \quad i = 1, 2, 3, \ldots\ldots, 47 \text{ and } m = 1, 2, \ldots, 6
\tag{8}
$$

where $T_{January}\left(P_{(mi)(Skardu)}\right)$ indicates the probabilities (the temporal weights) for varying drought categories in January of Skardu station. The $i$ shows the specific month (say, "January of 1971, January of 1972, January of 1973 and so on till January of 2017") varying over the selected data set (from January 1971 to December 2017). And $m$ denotes the

drought categories that are selected for the analysis (say, $m$ = ("1 (Extremely Dry (ED)), 2 (Severely Dry (SD)), 3 (Median dry (MD)), 4 (Normal Dry (ND)), 5 (Median Wet (MW)), 6 (Severely Wet (MW)), and 7 (extremely Wet (EW)))"). The varying drought categories are described in *Niaz et al. (2020)*. The steady-state weights for various drought categories for January of any year at the Skardu station are given by $W_{(mi)Skardu}$. The $\sum_{i=1}^{n} W_{(mi)Skardu}$ shows that the steady-state weights are added for all January at Skardu station with several drought categories. Moreover, $n$ show the total months of January (*i.e.*, 47) in Skardu. For instance, the nominator term $W_{(mi)Skardu}$ is computed with several drought categories for Skardu station for the various months of January, and the denominator term contains drought categories that appeared in all January at Skardu station. Now, the monthly weights for other months (February, up to December) with these selected categories are evaluated on the same rationale. Furthermore, the second phase perceives spatiotemporal characteristics of the selected drought categories. Thus, the spatiotemporal weights for these drought categories can be obtained as follows,

$$ST_{January}\left(P_{(mi)(Skardu)}\right) = \frac{T_{January}\left(P_{(mi)(Skardu)}\right)}{\sum_{j=1}^{M} Q_{mij}}, \quad i = 1, 2, 3, \ldots, 47 \text{ and } j = 1, 2, \ldots, 6 \ (9)$$

where Eq. (9) takes monthly spatiotemporal weights for various drought categories at Skardu station. $ST_{January}\left(P_{(mi)(Skardu)}\right)$ Shows the probabilities (spatiotemporal weights) computed from spatiotemporal information for varying drought categories in January at Skardu station. Further, the weights $T_{January}\left(P_{(mi)(Skardu)}\right)$ which were calculated from Eq. (8), are being further divided by the $\sum_{j=1}^{M} Q_{mij}$. Here, the quantity $Q_{mij}$ can be obtained for varying January of the selected period by adding varying drought categories ($m$) observed at various selected stations ($j$) and the total number of selected stations are denoted by $M$ (*i.e.*, $M$ = 6). The STTSSWS uses spatiotemporal characteristics of the selected stations and provides more accurate information about drought occurrences in a homogenous region. The obtained information from the STTSSWS can be used to build substantial drought monitoring procedures, techniques, and methodologies.

## The proposed MSTTSSWI method for selecting regional drought characteristics

The six homogenous stations are selected for the validation of the proposed procedures. The STTSSWS assigns weights to the various drought categories in each station. Further, the current study extends the concept of *Niaz et al. (2020)*. The mentioned study proposed a regional drought index at a 1-month time scale that selects suitable drought categories from the various homogenous stations. They used steady-state probabilities as a weighting scheme since they are associated with temporal information and are not considered spatial characteristics. Therefore, in the current study, the STTSSWS is used to obtain spatiotemporal information of the region. The obtained information from the STTSSWS is used to calculate suitable drought categories of the region. The vector of varying drought categories for regional characterization obtained from STTSSWS is known as the Maximum Spatio-Temporal Two-Stage Standardized Weighted Index

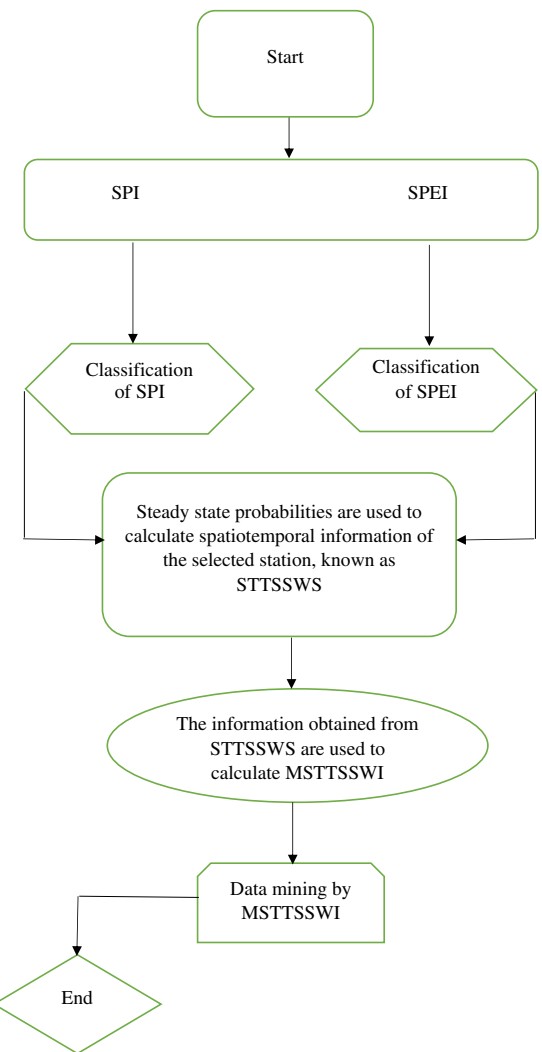

**Figure 1 Flowchart.** Flowchart for the proposed MSTTSSWI.

(MSTTSSWI) (Fig. 1). The mathematical form of the MSTTSSWI can be presented for SPEI at a 1-month time scale for varying stations as follows,

$$
\mathrm{MSTTSSWI} = \begin{cases}
SPEI\ Skardu\ if\ \prod_i (Skardu) > \prod_i (Gilgit) > \prod_i (Chilas) > \prod_i (Gupis) > \prod_i Bunji > \prod_i (Astore) \\
SPEI\ Gilgit\ if\ \prod_i (Gilgit) > \prod_i (Chilas) > \prod_i (Gupis) > \prod_i (Bunji) > \prod_i (Astore) \\
SPEI\ Chilas\ if\ \prod_i (Chilas) > \prod_i (Gupis) > \prod_i (Bunji) > \prod_i (Astore) \\
SPEI\ Gupuis\ if\ \prod_i (Gupis) > \prod_i (Bunji) > \prod_i (Astore) \\
SPEI\ Bunji\ if\ \prod_i (Gilgit) > \prod_i (Astore) \\
SPEI\ Astore,\ otherwise
\end{cases}
\tag{10}
$$

The MSTTSSWI is given in Eq. (10) for six selected stations. Each station may have varying drought categories for the selected time (from January 1971 to December 2017).
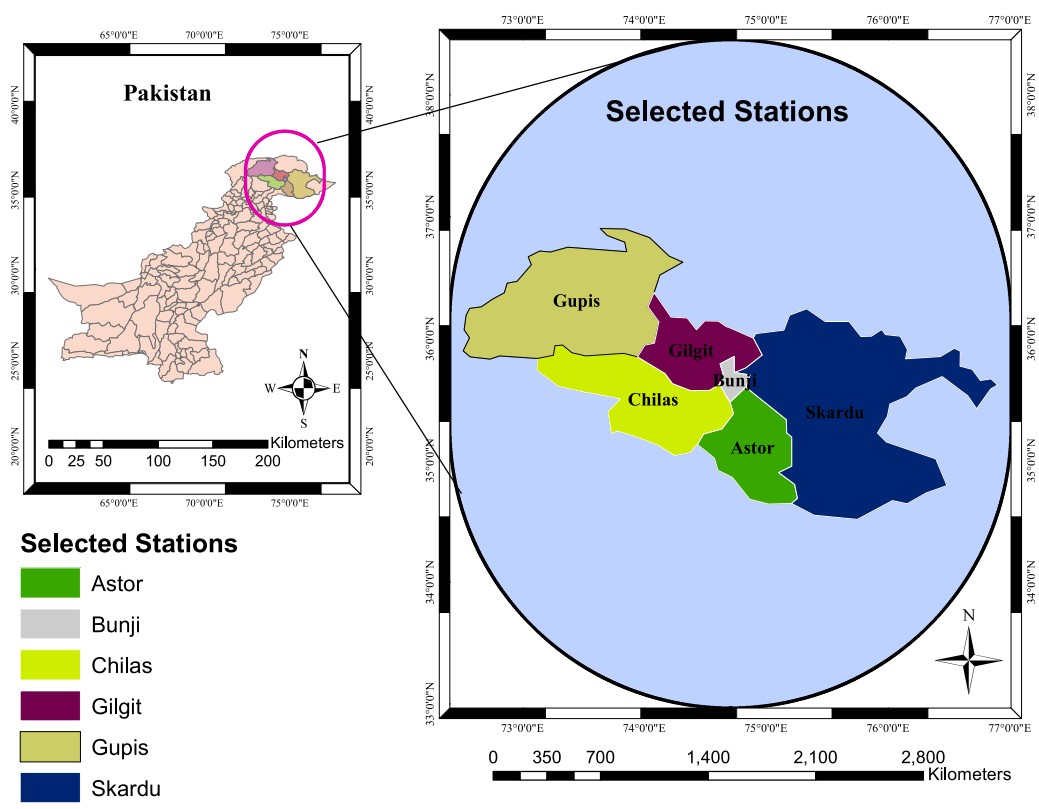

**Figure 2** **Geographical locations of the selected stations.**

The STTSSWS is used to assign weights for varying drought categories and stations. However, among the weights assigned to each selected station, the stations that receive maximum weights are selected in MSTTSSWI. Therefore, the vector of MSTTSSWI contains all suitable drought categories that are observed in various stations. For example, In January 1974, using SPI at a 1-month time scale, the STTSSWS assigns weights for varying drought categories as median wet (MW) by 0.06, (MW = 0.06), (MW = 0.07), normal dry (ND = 0.38), (ND = 0.41), and severely wet (SW = 0.03) observed at Skardu, Gilgit, Chilas, Gupis, Bunji, and Astore, respectively. In this case, for January 1974, the MSTTSSWI chooses the ND = 0.41, which is the maximum weight obtained from STTSSWS among various categories and stations. Accordingly, the selection is made for suitable drought categories for other months using SDI (SPI and SPEI). Conclusively, the MSTTSSWI provides the single vector of various appropriate drought categories among the six selected stations.

## APPLICATION

The six meteorological stations of the northern areas of Pakistan (Fig. 2) are selected in STTSSWS for the regional drought analysis. Northern Area is a geographic area with three mountain ranges, the Himalayas, Karakoram, and the Hindu Kush, which cover most of the region (*Rasul et al., 2011*). Many of the world's tallest peaks are found in this region, including K-2, Nanga Parbat, and Rakaposhi. The average altitude of Karakorum is

Peer

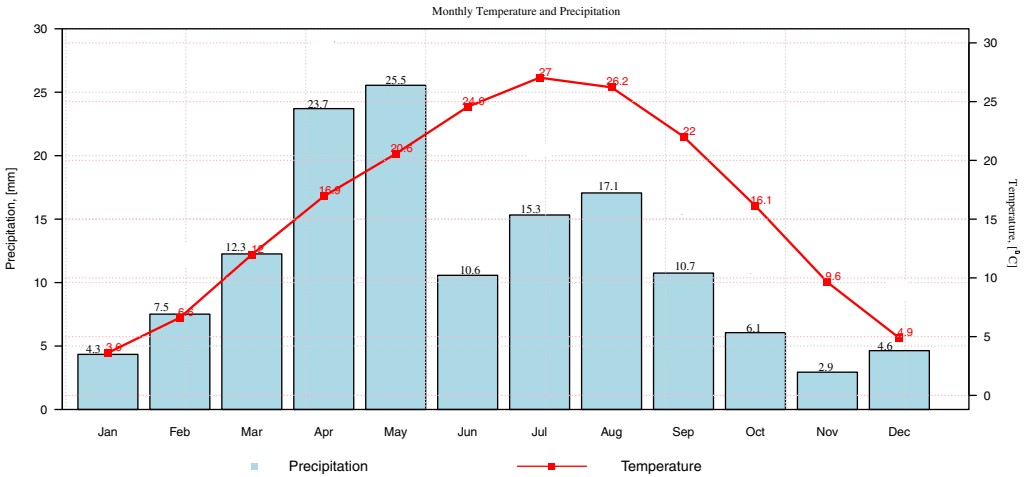

**Figure 3 The climograph.** The monthly precipitation and temperature are presented for Gilgit station.

(6,100 M), Hindukush (7,690 M) and Himalaya (8,848 M) (*Latif et al., 2020*). These high altitudes of mountains frequently deliver a significant portion of precipitation (*Rasul et al., 2011*; *Bocchiola & Diolaiuti, 2013*; *Adnan et al., 2017*). Further, this region's precipitation and temperature have substantial effects on the country's other regions (*Anjum et al., 2010*; *Bocchiola & Diolaiuti, 2013*; *Mazhar et al., 2020*; *Adnan et al., 2017*). Therefore, the precipitation and temperature of the selected region are used in STTSSWS to substantiate drought occurrences. In addition, the obtained information from STTSSWS is used to calculate MSTTSSWI that provides a regional characterization of meteorological drought.

## Results

The monthly data of precipitation, maximum and minimum temperature are observed in various stations. The observed data of these indicators (precipitation and temperature) are used for the current analysis. The monthly mean precipitation and mean monthly temperature (maximum and minimum temperature) for Gilgit station is presented in a climograph (Fig. 3). To avoid the presentation of multiple figures the data of Gilgit station is used for the climogrpah, however, the behavior of the selected indicators can be presented for other selected stations (Bunji, Gupis, Chilas, Skardu and Astore) accordingly. Further, based on the climatic conditions of the selected stations, two standardized drought indices are selected for drought classification. The standardization of these drought indices is done by using varying probability distributions. The distributions which are suitable according to climatic conditions are chosen for the standardizations. The Bayesian Information Criterion (BIC) is used to select appropriate probability distributions. In Table 1, at a one-month time scale, the 3p Weibull distribution shows suitable candidacy for the Astor station. The BIC of 3p Weibull distribution is −1036.5which is minimum among other distributions. Therefore, the distribution is used for the standardization in this station. The 3p Weibull distribution, at a 1-month time scale, shows suitable candidacy for Bunji station with BIC (−1,031.0), Gilgit with BIC

**Table 1 The BIC of various probability distributions.**

| index | Astore | | Bunji | | Gupis | |
| | Distribution | BIC | Distribution | BIC | Distribution | BIC |
|---|---|---|---|---|---|---|
| SPI | 3p Weibull | −1,036.5 | 3p Weibull | −1,031.0 | 4p Beta | −788.7 |
| SPEI | Trapezoidal | −710.1 | Johnson SB | −1,248.4 | Johnson SB | −977.6 |
| index | Chilas | | Gilgit | | Skardu | |
| | Distribution | BIC | Distribution | BIC | Distribution | BIC |
| SPI | 4P Beta | −805.6 | 3P Weibull | −1,097.4 | 3P Weibull | −735.1 |
| SPEI | Johnson SB | −594.7 | Johnson SB | −1,213.2 | Trapezoidal | −664.6 |

Note:
The BIC of various probability distributions for selected stations at scale-1 for SPI and SPEI.

**Table 2 The preliminary classification.** The preliminary classification of various drought categories based on SDI.

| SDI | Major drought classes |
|---|---|
| SDI >= 2 | Extremely Wet (EW) |
| SDI > 1.5 & SDI <= 2 | Severely (SW) |
| SDI > 1 & SDI <= 1.5 | Median Wet (MW) |
| SDI > −1 & SDI <= 1 | Normal Dry (ND) |
| SDI > −1.5 1 & SDI <= −1 | Median Dry (MD) |
| SDI > −2 & SDI <= −1.5 | Severely Dry (SD) |
| SDI >= −2 | Extremely Dry (ED) |

(−1,097), and for Skardu with BIC (−735.1). The 4p Beta distribution shows better candidacy at a 1-month SPI for two stations, including Gupis and Chilas with BIC −788.7 and −805.6, respectively. Further, for SPEI at a 1-month time scale, the Trapezoidal distribution is fitting suitably for station Astor and Skardu with BIC −710.1 and −664.6, respectively. In Bunji, Gupis, Chilas, and Gilgit, the Johnson SB distribution is a suitable candidate concerning their minimum BIC values, accordingly. After the standardization of the data, the observed values are classified for various drought categories by using SPI and SPEI. The drought classification shows the multiple levels of drought categories (*Li et al., 2015*). For instance, the SDI (SPI and SPEI) value less than or equal to −2 represents the extremely dry and greater than two classified as extreme wet conditions and so forth (Table 2).

Furthermore, Fig. 4 shows theoretical and empirical distributions for SPI at a 1-month time scale (SPI-1), and theoretical and empirical distributions for SPEI at a 1-month time scale (SPEI-1) for various stations are presented in Fig. 5. Figure 6 shows the temporal behavior of SPI-1 at selected stations. Further, the temporal behavior of SPEI-1 at selected stations can be observed in Fig. 7. These indices are used to find the drought occurrences in the selected stations. The drought occurrences based on SPI and SPEI in various stations are observed accordingly. The monthly drought occurrence for the year of 2017, obtained from the SPI and SPEI are presented in Tables 3 and 4 respectively.

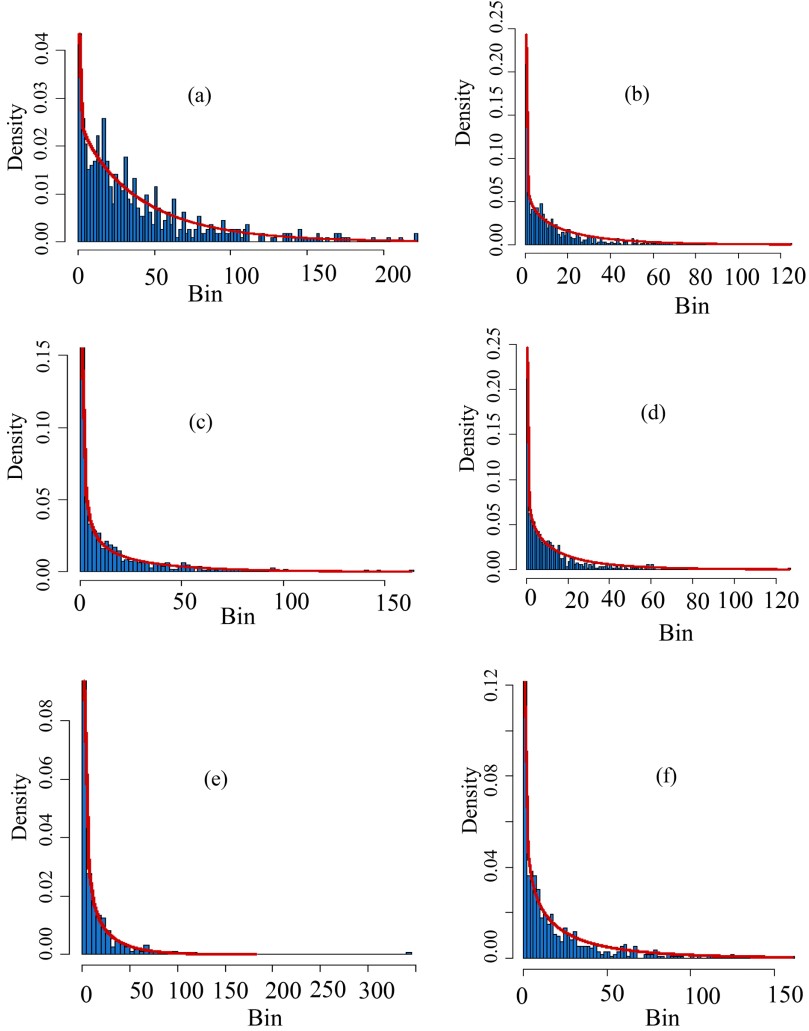

**Figure 4 Histograms of the selected distributions.** The theoretical *vs* empirical histograms are presented based on SPI at scale-1 for varying stations. For example, the theoretical *vs* empirical histograms for Astore station is presented in (A), for Bunji it is presented in (B), for Gupis it can be seen in (C), for Chilas , Giligit and Skardu, these are presented in (D–F) respectively. Further, in the multiple sections of the figure, the bins on the horizontal axis are used for ranges of data, and the ratio of the relative frequency of any specified bins' interval to its width size is denoted by density on the vertical axis. Moreover, from the fitted lines to the multiple sections it can be observed that for SPI-1, the Gupis and Gilgit data have more closeness between the theoretical and empirical.

It can be observed that in both SPI and SPEI computation the ND from the selected stations is the prevalent category. Hence, the ND category should be considered as an important category for further analysis. Further, information obtained from these indices is used for the computation of STTSSWS. The STTSSWS obtains spatiotemporal information for various stations. The obtained weights from SPI at a 1-month time scale are given in Table 5. The STTSSWS contains the temporal and spatial information of the whole region and provides more comprehensive and precise results for varying drought categories. For example, in January, for the Astore station at the 1-month time scale of SPI, the SD takes a value of 0.0268. The value shows that the SD has very less likely

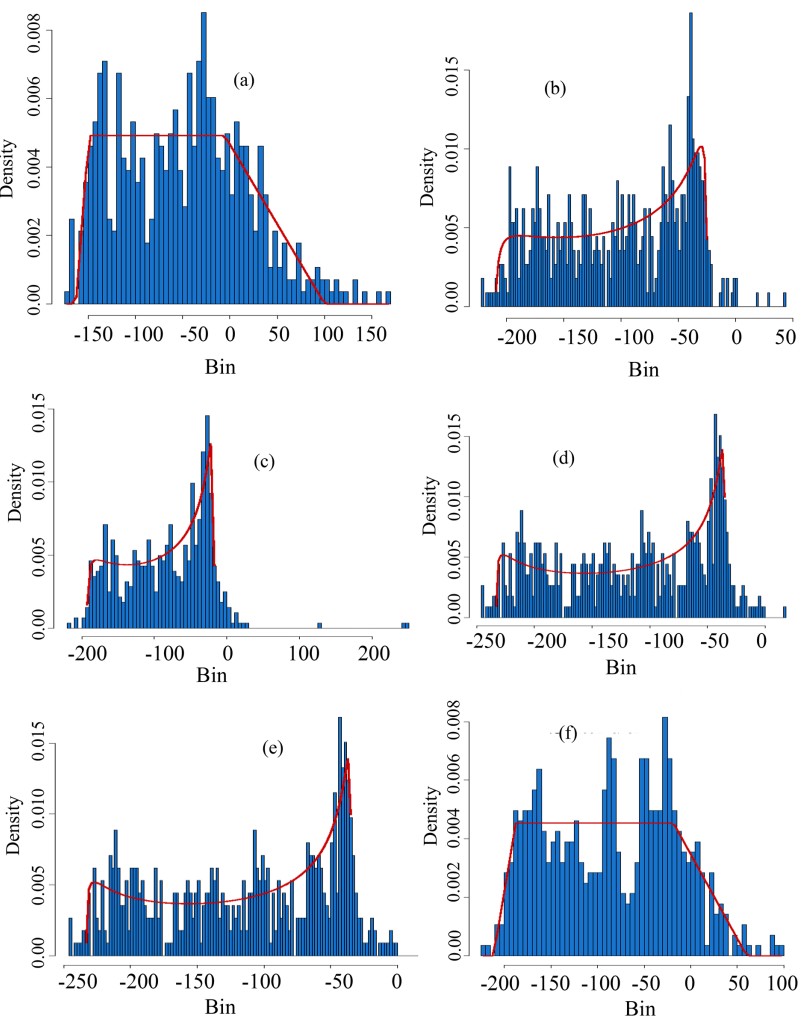

**Figure 5 Theoretical *vs* empirical histograms.** Theoretical *vs* empirical histograms of the selected distributions based on SPEI at scale-1 for selected stations are presented accordingly. The theoretical *vs* empirical histograms for Astore station is presented in (A), for Bunji it is presented in (B), for Gupis, Chilas, Giligit and Skardu, the theoretical *vs* empirical histograms are presented in (C–F), respectively.

to occur in January. However, in January, the ND is more likely to occur in Gilgit station among other stations of the region with the weight (0.2672). For other stations and months, the weights for varying drought categories can be observed. Furthermore, STTSSWS weights for a homogenous region using SPEI at a 1-month time scale are presented in Table 6. In the Astore station, at the 1-month time scale of SPEI, the MW takes a value of 0.0836. The value shows that the MW has significantly less likely to occur in January at Gilgit. However, in January, the ND is more likely to occur in the Skardu station among other region stations with the weight (0.1844). Moreover, to avoid the complexity in presenting results, we just presented results for the particular year, 2017. However, the results of selected years can be observed from the proposed scheme. Further, the STTSSWS is used to assign weights for the selected drought categories on varying stations. The STTSSWS is applied to the six meteorological stations to obtain
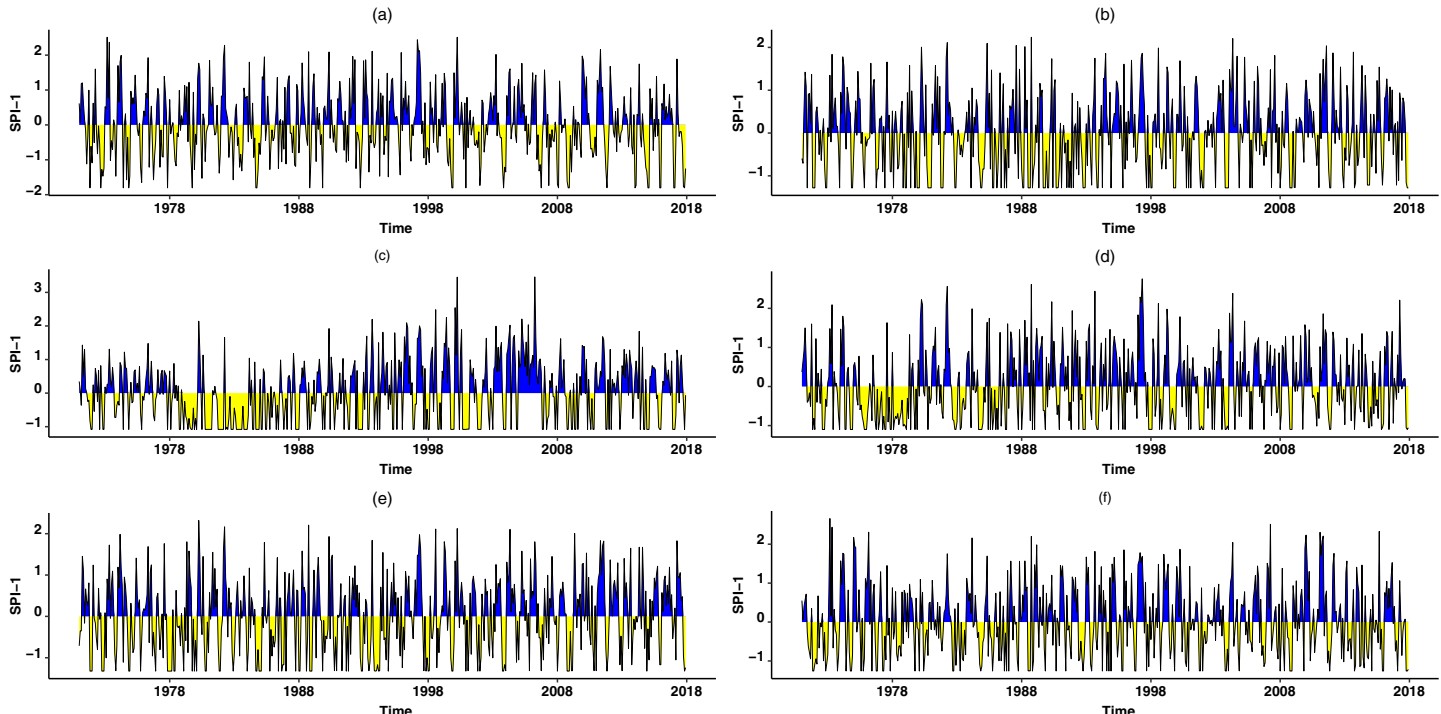

**Figure 6 Temporal behaviour.** Temporal behavior in various plots of the selected stations for SPI at scale-1 (SPI-1). The temporal behavior for SPI-1 at Astore station can be observed in (A). (B) Temporal behavior of SPI-1 in Bunji. (C) The SPI-1 temporal behavior is presented for Gupis. Moreover, the temporal behavior of SPI-1 for Chilas, Gilgit, and Skardu can be observed from (D–F), respectively.

spatiotemporal weights for estimating MSTTSSWI. Hence, the MSTTSSWI provides more comprehensive and accurate information for the regional drought characteristics. The behavior of the proposed MSTTSSWI can be observed by (Fig. 8).

## Discussion

The two drought indices (SPI & SPEI) are considered in the current analysis. These selected drought indices provide the standardized values for the given climate indicators (precipitation and temperature) in the selected stations. The appropriate probability distributions according to time scales and stations are selected for the standardization (*Niaz et al., 2020*; *Ali et al., 2020*; *Niaz et al., 2021*; *Raza et al., 2021*). The BIC criteria are used to select these probability distributions. Further, the steady-state probabilities are used for the computation of STTSSWS. The STTSSWS works in two stages; in the first stage, SPI, SPEI, and the steady-state probabilities are calculated for each station separately. The steady-state probabilities consider temporal information of the stations regardless of the spatial accountability of the region. However, the current study aimed to improve the information of the regional drought characteristics. Therefore, in stage two of the STTSSWS, the steady-state probabilities are used to calculate spatiotemporal information (weights) for the varying drought categories. Hence, the proposed scheme used spatial and temporal characteristics of regional drought to calculate weights for the various drought categories. Various studies emphasized calculating spatiotemporal information of the

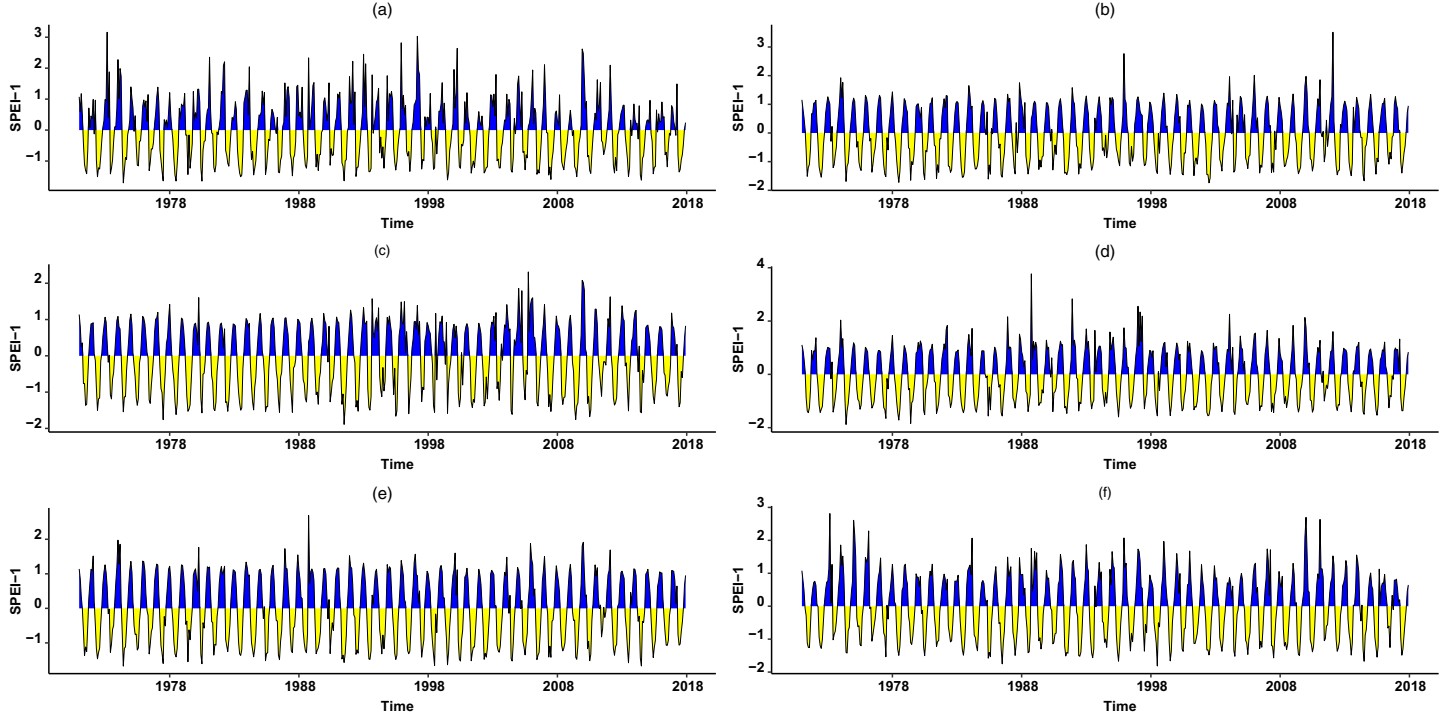

**Figure 7 Temporal behavior in various plots for SPEI.** Temporal behavior of SPEI at scale-1 (SPEI-1) can be observed in varying plots for the selected stations. The temporal behavior for SPEI-1 at Astore station can be seen in (A). (B) Temporal behavior of SPEI-1 is presented for Bunji. The temporal behavior of SPEI-1 in Gupis is presented in (C). Further, the temporal behavior of SPEI-1 for Chilas, Gilgit, and Skardu can be observed from (D–F), respectively.

**Table 3 The classification of the drought categories.**

|  | Astore | | Bunji | | Gupis | | Chilas | | Gilgit | | Skardu | |
|---|---|---|---|---|---|---|---|---|---|---|---|---|
|  | Index | Classif. | Index | Classif. | Index | Classif. | Index | Classif. | Index | Classif. | Index | Classif. |
| January | −1.8044 | SD | −1.2152 | MD | −1.0805 | MD | −0.3254 | ND | −0.2296 | ND | −0.4827 | ND |
| February | −1.8044 | SD | 0.2629 | ND | −1.0805 | MD | 0.8075 | ND | 0.2014 | ND | −1.2581 | ND |
| March | −0.4745 | ND | −1.1059 | MD | −0.6655 | ND | −0.1535 | ND | −1.0523 | MD | −0.3096 | ND |
| April | 1.8879 | SW | 0.9377 | ND | 1.2811 | MW | 2.2094 | EW | 1.8283 | SW | 1.0615 | EW |
| May | 0.3314 | ND | 0.6975 | ND | 0.9813 | ND | 0.8297 | ND | 0.9724 | ND | 0.0796 | ND |
| June | −0.3302 | ND | −0.6380 | ND | −0.2883 | ND | −0.0637 | ND | 0.9309 | ND | −0.8574 | ND |
| July | −0.1857 | ND | 0.8192 | ND | 0.7872 | ND | 0.1366 | ND | 1.0574 | MW | −0.2401 | ND |
| August | −0.1967 | ND | 0.7181 | ND | 1.1317 | MW | 0.2058 | ND | 0.3864 | ND | 0.0086 | ND |
| September | −0.5938 | ND | 0.2824 | ND | 0.4029 | ND | 0.1937 | ND | 0.4759 | ND | 0.0739 | ND |
| October | −1.7685 | SD | −1.1576 | MD | −0.2883 | ND | −1.0539 | MD | −0.9569 | ND | −1.2581 | MD |
| November | −1.8044 | SD | −1.2806 | MD | −1.0805 | MD | −1.0995 | MD | −1.3227 | MD | −1.2581 | MD |
| December | −1.2499 | MD | −1.2806 | MD | −0.0621 | ND | −1.0539 | MD | −1.2398 | MD | −1.2168 | MD |

**Note:**
The classified (Classif.) based on SPI. The varying drought categories observed in various months of the year 2017, in selected stations.

**Table 4 The classification based on SPEI.**

| | Astore | | Bunji | | Gupis | | Chilas | | Gilgit | | Skardu | |
|---|---|---|---|---|---|---|---|---|---|---|---|---|
| | Index | Classif. | Index | Classif. | Index | Classif. | Index | Classif. | Index | Classif. | Index | Classif. |
| January | 0.2741 | ND | 0.9796 | ND | 0.7679 | ND | 0.9039 | ND | 1.0670 | MW | 0.8220 | ND |
| February | 0.0829 | ND | 0.8384 | ND | 0.4871 | ND | 0.9296 | ND | 0.7976 | ND | 0.4919 | ND |
| March | −0.1682 | ND | 0.1045 | ND | 0.0219 | ND | 0.1193 | ND | 0.1447 | ND | 0.1232 | ND |
| April | 1.4903 | MW | 0.0887 | ND | 0.3163 | ND | 1.3240 | MW | 0.6436 | ND | 0.1957 | ND |
| May | −0.8214 | ND | −1.0027 | MD | −0.7708 | ND | −0.8617 | ND | −0.7646 | ND | −0.9650 | ND |
| June | −1.3547 | MD | −1.3982 | MD | −1.4032 | MD | −1.3619 | MD | −1.0659 | MD | −1.4873 | MD |
| July | −1.2755 | MD | −1.0733 | MD | −1.2931 | MD | −1.3665 | MD | −1.0312 | MD | −1.3089 | MD |
| August | −0.9560 | ND | −0.7027 | ND | −0.4335 | ND | −0.8499 | ND | −0.6704 | ND | −0.8895 | ND |
| September | −0.7888 | ND | −0.4535 | ND | −0.5996 | ND | −0.4873 | ND | −0.3957 | ND | −0.4748 | ND |
| October | −0.4409 | ND | 0.0387 | ND | −0.0183 | ND | 0.0061 | ND | 0.0157 | ND | −0.0880 | ND |
| November | 0.0596 | ND | 0.7154 | ND | 0.5016 | ND | 0.6265 | ND | 0.6878 | ND | 0.4532 | ND |
| December | 0.2411 | ND | 0.9475 | ND | 0.8261 | ND | 0.8367 | ND | 0.9538 | ND | 0.6461 | ND |

Note:
The classified (Classif.) drought categories observed in various months for SPEI at a 1 month-time scale of the year 2017, in selected stations.

**Table 5 The weights obtained from STTSSWS.**

**SPI-1**

| Station | Astore | | Bunji | | Gupis | | Chilas | | Gilgit | | Skardu | | |
|---|---|---|---|---|---|---|---|---|---|---|---|---|---|
| Month | Category | Weight | Category | Weight | Category | Weight | Category | Weight | Category | Weight | Category | Weight | Sum |
| January | SD | 0.0268 | MD | 0.0750 | MD | 0.1106 | ND | 0.2566 | ND | 0.2672 | ND | 0.2638 | 1 |
| February | SD | 0.0233 | ND | 0.2126 | MD | 0.1068 | ND | 0.2175 | ND | 0.2179 | ND | 0.2219 | 1 |
| March | ND | 0.2397 | MD | 0.0545 | ND | 0.1898 | ND | 0.2295 | MD | 0.0541 | ND | 0.2325 | 1 |
| April | SW | 0.0606 | ND | 0.7149 | MW | 0.1037 | EW | 0.0333 | SW | 0.0582 | EW | 0.0294 | 1 |
| May | ND | 0.1646 | ND | 0.1769 | ND | 0.1610 | ND | 0.1571 | ND | 0.1817 | ND | 0.1587 | 1 |
| June | ND | 0.1609 | ND | 0.1637 | ND | 0.1959 | ND | 0.1626 | ND | 0.1518 | ND | 0.1653 | 1 |
| July | ND | 0.1789 | ND | 0.2072 | ND | 0.1978 | ND | 0.1948 | MW | 0.0241 | ND | 0.1971 | 1 |
| August | ND | 0.1794 | ND | 0.2221 | MW | 0.0314 | ND | 0.1823 | ND | 0.2014 | ND | 0.1833 | 1 |
| September | ND | 0.1815 | ND | 0.1848 | ND | 0.1624 | ND | 0.1554 | ND | 0.1592 | ND | 0.1568 | 1 |
| October | SD | 0.0372 | MD | 0.1053 | ND | 0.3256 | MD | 0.1074 | ND | 0.3396 | MD | 0.0849 | 1 |
| November | SD | 0.1815 | MD | 0.1848 | MD | 0.1624 | MD | 0.1554 | MD | 0.1592 | MD | 0.1568 | 1 |
| December | MD | 0.0600 | MD | 0.1342 | ND | 0.3846 | MD | 0.1511 | MD | 0.1481 | MD | 0.1221 | 1 |

Note:
The weights obtained from STTSSWS are provided for the year 2017. These weights are calculated by SPI and steady-state probabilities.

drought (*Corzo Perez et al., 2011*; *Wang et al., 2020*; *Diaz et al., 2020a*). The knowledge about spatiotemporal characteristics of the drought can help for accurate drought monitoring characteristics and can be used for significant modeling and drought prediction (*Caloiero et al., 2018*; *Zhou et al., 2020*). Therefore, the current study aimed to develop a new drought assessment procedure for the characterization of regional drought. The MSTTSSWI uses STTSSWS as a weighing scheme to provide more comprehensive and accurate information about the regional drought characteristics.

**Table 6 The weights obtained from STTSSWS are given for the year 2017 using SPEI and steady-state probabilities.**

**SPEI-1**

| Station | Astore | | Bunji | | Gupis | | Chilas | | Gilgit | | Skardu | | |
|---------|----------|--------|----------|--------|----------|--------|----------|--------|----------|--------|----------|--------|-----|
| Month | Category | Weight | Category | Weight | Category | Weight | Category | Weight | Category | Weight | Category | Weight | Sum |
| January | ND | 0.1379 | ND | 0.2569 | ND | 0.1697 | ND | 0.1676 | MW | 0.0836 | ND | 0.1844 | 1 |
| February | ND | 0.1672 | ND | 0.1651 | ND | 0.1391 | ND | 0.1771 | ND | 0.1661 | ND | 0.1854 | 1 |
| March | ND | 0.2147 | ND | 0.1435 | ND | 0.1471 | ND | 0.1637 | ND | 0.1411 | ND | 0.1898 | 1 |
| April | MW | 0.0359 | ND | 0.2276 | ND | 0.2348 | MW | 0.0401 | ND | 0.2322 | ND | 0.2295 | 1 |
| May | ND | 0.2056 | MD | 0.0616 | ND | 0.1878 | ND | 0.1900 | ND | 0.1768 | ND | 0.1781 | 1 |
| June | MD | 0.1146 | MD | 0.1925 | MD | 0.1429 | MD | 0.1899 | MD | 0.1952 | MD | 0.1649 | 1 |
| July | MD | 0.1509 | MD | 0.1359 | MD | 0.1898 | MD | 0.1668 | MD | 0.1827 | MD | 0.1739 | 1 |
| August | ND | 0.1688 | ND | 0.1619 | ND | 0.1693 | ND | 0.1525 | ND | 0.1610 | ND | 0.1866 | 1 |
| September | ND | 0.1852 | ND | 0.1815 | ND | 0.0816 | ND | 0.1853 | ND | 0.1815 | ND | 0.1848 | 1 |
| October | ND | 0.1644 | ND | 0.1645 | ND | 0.1645 | ND | 0.1644 | ND | 0.1779 | ND | 0.1643 | 1 |
| November | ND | 0.1720 | ND | 0.1642 | ND | 0.1618 | ND | 0.1719 | ND | 0.1655 | ND | 0.1647 | 1 |
| December | ND | 0.1087 | ND | 0.2373 | ND | 0.1276 | ND | 0.1371 | ND | 0.2683 | ND | 0.1211 | 1 |

**Note:**
The weights obtained from STTSSWS are given for the year 2017. These weights are calculated by SPEI and steady-state probabilities accordingly.

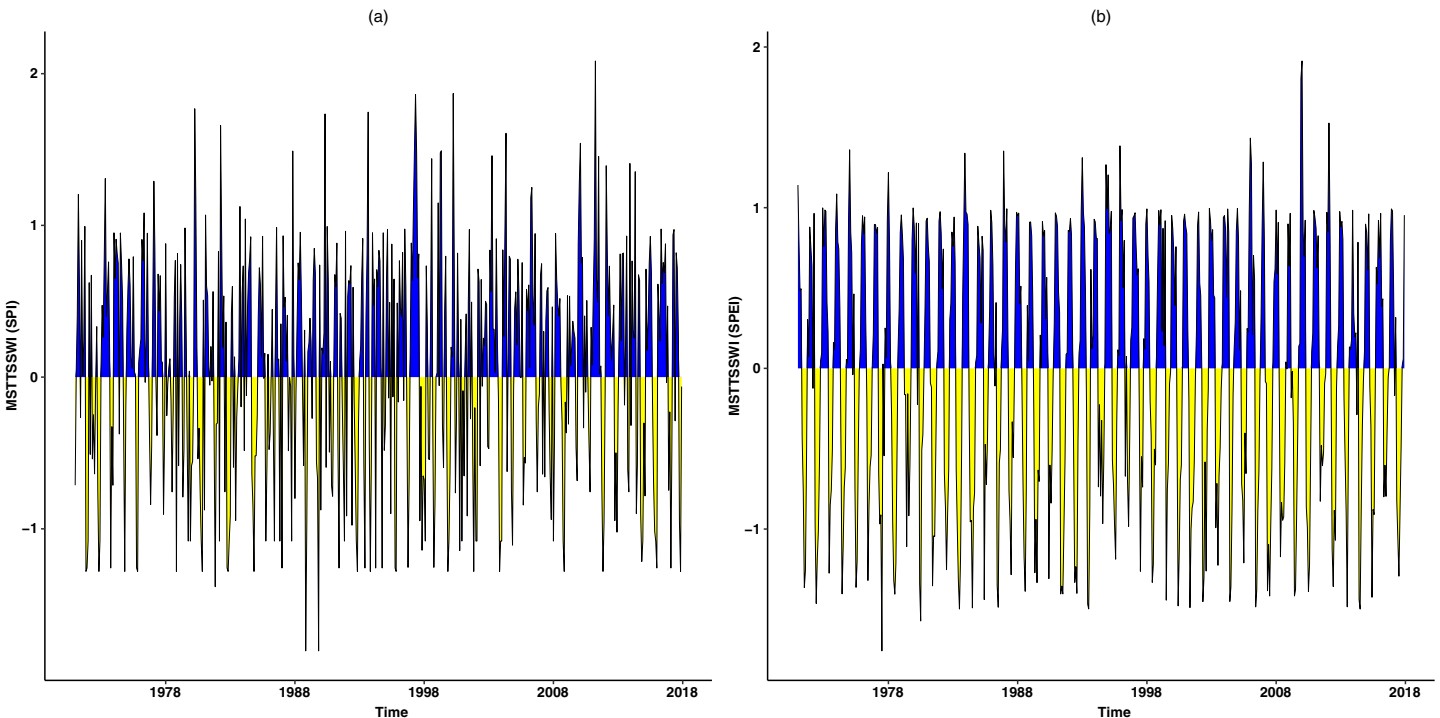

**Figure 8 The temporal behaviour of the MSTTSSWI.** The temporal behavior of the MSTTSSWI is presented based on SPI-1 and SPEI-1.

## CONCLUSION

The current study proposes a new drought assessment procedure, known as MSTTSSWI based on STTSSWS. The SPI, SPEI, and steady-state probabilities are used in STTSSWS to obtain new spatiotemporal weights for various drought categories. Further, the STTSSWS is used as a weighting scheme to calculate MSTTSSWI to obtain more accurate and precise spatiotemporal information about drought occurrences at the regional level. The outcomes of the proposed procedure MSTTSSWI provide regional spatiotemporal characteristics for the drought in the selected region and motivate researchers and policymakers to use the more comprehensive and accurate spatiotemporal characterization of drought in the selected region. Information obtained from MSTTSSWI can be applied for monitoring and forecasting drought more accurately. Moreover, when the climatic conditions of the stations change, the proposed MSTTSSWI works accordingly to the specific conditions.

### Funding
The authors were supported by the Deanship of Scientific Research at King Saud University, through research group no 1435-075. The funders had no role in study design, data collection and analysis, decision to publish, or preparation of the manuscript.

### Grant Disclosures
The following grant information was disclosed by the authors:
Deanship of Scientific Research at King Saud University: 1435-075.

### Competing Interests
The authors declare that they have no competing interests.

### Author Contributions
- Rizwan Niaz conceived and designed the experiments, performed the experiments, analyzed the data, prepared figures and/or tables, authored or reviewed drafts of the paper, and approved the final draft.
- Nouman Iqbal conceived and designed the experiments, analyzed the data, prepared figures and/or tables, and approved the final draft.
- Nadhir Al-Ansari conceived and designed the experiments, authored or reviewed drafts of the paper, and approved the final draft.
- Ijaz Hussain conceived and designed the experiments, prepared figures and/or tables, supervision, and approved the final draft.
- Elsayed Elsherbini Elashkar conceived and designed the experiments, performed the experiments, analyzed the data, authored or reviewed drafts of the paper, and approved the final draft.
- Sadaf Shamshoddin Soudagar conceived and designed the experiments, performed the experiments, prepared figures and/or tables, and approved the final draft.

- Showkat Hussain Gani conceived and designed the experiments, performed the experiments, analyzed the data, authored or reviewed drafts of the paper, and approved the final draft.
- Alaa Mohamd Shoukry conceived and designed the experiments, performed the experiments, prepared figures and/or tables, authored or reviewed drafts of the paper, and approved the final draft.
- Saad Sh. Sammen performed the experiments, analyzed the data, prepared figures and/or tables, and approved the final draft.

## Data Availability

The data is available in the Supplemental File.

## Supplemental Information

Supplemental information for this article can be found online at http://dx.doi.org/10.7717/peerj.13249#supplemental-information.

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
