# Peer review of "A new spatiotemporal two-stage standardized weighted procedure for regional drought analysis"

_PeerJ, doi:10.7717/peerj.13249_

## Round 0.1 · original submission · Major Revisions

Three referees have commented on your article. From their reports, there is clear merit to the article but also some important concerns. Indeed one issue that is highlighted by the referees, especially in the reports from referees #2 and #3, is a set of concerns with the weights. Given that these are central to the method and an important feature throughout the discussion the issues connected with the weighting require attention. There is also a need to substantially enhance aspects such as the illustrations. Please note that major revision is required (minor revision will be inadequate) and I suggest that you pay particular regard to the comments of referee #3.

Reviewer 1 ·

Basic reporting

The current effort with the title “A novel spatio-temporal two-stage standardized
weighting scheme for regional drought analysis” applies a composite indicator (SDI), which combines meteorological (SPI) and Agricultural (SPEI) aspects. The article is well structured and technically correct in general, with the overarching aim of showing that the Index can be used in areas with similar conditions. In addition, there is an issue with the methodology. It is underlined, as also pertinent research has recommended, to avoid time scales of 3-month or shorter for the indices estimation, since Pakistan exhibits characteristics with extremely low precipitation during summers. Thus, it is suggested to apply the time scales of 6 and 12 months to characterize more accurately droughts and the analysis results to be comparable across the continent.

Experimental design

- Please improve the introduction with the following terms composite indicators, drought indicators, and the statistical weighting methods (e.g. OECD, European Commission (ed) (2008) Handbook on constructing composite indicators: methodology and user guide. Organisation for Economic Co-operation and Development Publishing, Paris).
- It is proposed to use a hill shade background with transparency to improve the quality of the map. In addition, it will be helpful for the potential readers a base map of the whole area.

Validity of the findings

The quality of the figures and maps should be improved.

Reviewer 2 ·

Basic reporting

.

Experimental design

.

Validity of the findings

.

Additional comments

Manuscript No: #65175

Title: A novel spatio-temporal two-stage standardized weighting scheme for regional drought analysis.

1. Pg. 8: Line 82-83: How has this index been validated. Plz explain in detail.

2. Pg.10: Line 112: Change "Moreover, in a Markov process, it can be more explicitly defined as the probabilities are approached the steady-state probabilities .......

Change to:

Moreover, in a Markov process, it can be more explicitly defined as the probabilities approach the steady-state probabilities.......

3. Pg 13: Line 200-202: Multiple distributions have been used to the one-month SPI and one-month SPEI at different sites. How has it been ascertained that these scheme is statistically consistent ?

4. Pg 13: Line 200-202: How has the weights obtained from STTSSWS been used to derive the various drought classes ?

5. Pg. 14 Line 200-212: What could be the reason that one month-SPI and one-month SPEI has been used in this analysis ? Why the 3-month, 6-month and 12-month timescale SPI's not used in the analysis ?

6. Has the indicators levels (weights) used for deciding the various classes of drought levels been fine-tuned with the actual field conditions during drought periods ?

7. How sensitive are these weights for various drought levels ?

8. The resolution of all the figures needs improvement.

9. As compared to Figure 1, the study area depicted in Figure 7 & 8 seems to be elongated and is not in proportion (to scale). This may be rectified.

Reviewer 3 ·

Basic reporting

The authors propose a method to ponderate the drought classes obtained from standardized drought indices.

Improving methods, proposing indices, and improving approaches to monitoring drought is important for its better management. However, the document has several methodological shortcomings that make it unsuitable for publication. The paper structure needs an improvement too.

Experimental design

-The methodology is not clear. The approach makes use of two meteorological drought indices (SPI and SPEI), their methodology must be included, even in a general way, since the method to weight the type of classes is based on this type of indices.
-It is difficult for readers to keep track of how the weights are calculated.
-It is also necessary to describe the type of drought classes according to the type of index used.
-The methodology for calculating STTSSWS is not clear. I suggest describing the steps that are necessary to calculate it.
-What is the justification for weighting by drought classes?
-Why not directly use the values ​​of the indices to calculate the weights?
-What is the difference between this method and the use of a threshold directly applied to the values ​​of the hydrometeorological variable?
-In a practical case, what is the significance of the weights obtained by the proposed method?
-In general, the occurrence of the Extreme class is less than the Severe, which is, in turn, less than the Moderate one (Median as mentioned in the document), why then weight by classes?
-Why not monitor drought directly by type of drought (i.e. Extreme, Severe, Moderate)?

Validity of the findings

The Discussion section requires improvement.
Some points to discuss are:
-The advantages of the method (compared to other approaches)
-Disadvantages, see the previous section.
-Since this paper is more for drought monitoring, it would be interesting to provide the reader with a guide to interpret the results obtained.
-Difference / Similarity with respect to other approaches.
-Difference / Similarity with respect to other studies applied in the study area.

Additional comments

I'm not sure if this method is novel, the use of drought classes (Extreme, Severe, and Moderate) is widely used and discussed, and also the treatment of this type of information.

Introduction Section.
What about the methods that address drought as an object that changes in space and time?
Corzo Perez et al. (2011), Diaz et al. (2020a, 2020b), van Huijgevoort et al. (2013), Vernieuwe et al. (2020) and Zscheischler et al. (2013)

It is not clear the motivation.


Line 61 “drought monitoring”?
Lines 65-69 The sentences are not clear. SPEI uses also PET
Lines 71 What issues? Indicate some of them.
Line 73 what does mean complicated phenomenon?
Line 73 what does mean complex occurrence?
Line 74 now, drought monitoring policies? What this term means
Line 2.1 Where is the standardized drought index? This is an introduction of SPI and SPEI, but the methodology is not described.
Line 155 what does mean imperative knowledge?
Line 103-157 STTSSWS methodology is not clear. It’s difficult to follow the logic of the procedure.
Line 169 this was already shown in the Introduction sect.
Line 171-772 this is not the place for this sentence.
In Application (I assume is the Experimental setup), the experiment has to be described, what specifications other colleagues have to follow in order to reproduce your experiment.
Line 184 SDI? How this drought index was calculated?
Line 189-194 Where this procedure was described? Where Weibull, 4p Beta Johnson SB distribution were described?
Line 235 BIC criteria?
Line 241-242 How “the class” reflects for consideration in the region?
Line 242-246 this is not the place for this text. This is not a discussion.
Line 246 It is not clear how the introduced index or method provides more appropriate spatiotemporal information. What do you mean by “appropriate spatiotemporal information”?
Line 253-257 this text is not part of your conclusions, this part is for the introduction.
Line 258-259 See comments on Line 242-246


References
Corzo Perez, G.A., van Huijgevoort, M.H.J., Voß, F., and van Lanen, H.A.J. (2011). On the spatio-temporal analysis of hydrological droughts from global hydrological models. Hydrology and Earth System Sciences, 15(9), 2963–2978. https://doi.org/10.5194/hess-15-2963-2011

Diaz, V., Corzo Perez, G.A., Van Lanen, H.A.J., Solomatine, D., and Varouchakis, E.A. (2020a). An approach to characterise spatio-temporal drought dynamics. Advances in Water Resources, 137(January), 103512. https://doi.org/10.1016/j.advwatres.2020.103512

Diaz, V., Corzo Perez, G.A., Van Lanen, H.A.J., Solomatine, D., and Varouchakis, E.A. (2020b). Characterisation of the dynamics of past droughts. Science of The Total Environment, 718, 134588. https://doi.org/10.1016/j.scitotenv.2019.134588

van Huijgevoort, M.H.J., Hazenberg, P., van Lanen, H.A.J., Teuling, A.J., Clark, D.B., Folwell, S., Gosling, S.N., Hanasaki, N., Heinke, J., Koirala, S., Stacke, T., Voss, F., Sheffield, J., and Uijlenhoet, R. (2013). Global Multimodel Analysis of Drought in Runoff for the Second Half of the Twentieth Century. Journal of Hydrometeorology, 14(5), 1535–1552. https://doi.org/10.1175/JHM-D-12-0186.1

Vernieuwe, H., De Baets, B., and Verhoest, N.E.C. (2020). A mathematical morphology approach for a qualitative exploration of drought events in space and time. International Journal of Climatology, 40(1), 530–543. https://doi.org/10.1002/joc.6226

Zscheischler, J., Mahecha, M.D., Harmeling, S., and Reichstein, M. (2013). Detection and attribution of large spatiotemporal extreme events in Earth observation data. Ecological Informatics, 15, 66–73. https://doi.org/10.1016/j.ecoinf.2013.03.004

---

## Round 0.2 · Minor Revisions

The revisions made appear to have enhanced the article but there are some issues that require further attention:

1. There appear to have been some changes to authorship. Notably the insertion of Al-Ansari and removal of Zhang. Can you please explain the authorship changes (especially of Zhang) and provide assurance that all authors and Zhang are in agreement with the authorship listed on the next version of the manuscript.

2. There are a very large number of Figures and some do not seem necessary or may convey material that could be better presented in tabular form.

3. Colour is often used in an unclear way. For example, in Figs 3-5 (just as examples - the comment also applies elsewhere) there appears to be no logic to the colours (e.g. there is no colour scale, just colour used to identify different items). This is confusing when there is a colour scale and then a better set of colour might be used. Surely this material would be better presented in Tables?

4. In other Figures there is a need to explain the colours used (e.g. Fig 7 - colour bar and lines need explanation). Also in all multi-component Figs it is normal to label the component parts (a), (b)...etc and to explain the content in the caption for (a), (b)...etc.

5. What is the value of Fig 11a when all the other components have location names indicated? Note in Fig 11 the colours used are a little confusing as there is no evident scale and hence the relative magnitude of two colours may differ between components (e.g. blue and purple in (c) and (d)). Again, this (and other Figures), may be better presented as Tables.

---

## Round 0.3 · Minor Revisions

Thank you for revising the manuscript and the helpful document replying to the last decision message. Although enhanced, there are still concerns with some of the Figures. In particular, Figures 3-5 and 10-11 are a little awkward. The content of these Figures would be easier to present in tabular form. Could you convert these Figures into Tables (unless there is a powerful argument to keep as Figures).

---

## Round 0.4 · accepted · Accept

Thank you for revising your article, it looks much stronger as a result.